# High-Affinity Plasma Membrane Ca^2+^ Channel Cch1 Modulates Adaptation to Sodium Dodecyl Sulfate-Triggered Rise in Cytosolic Ca^2+^ Concentration in *Ogataea parapolymorpha*

**DOI:** 10.3390/ijms252111450

**Published:** 2024-10-25

**Authors:** Maria Kulakova, Maria Pakhomova, Victoria Bidiuk, Alexey Ershov, Alexander Alexandrov, Michael Agaphonov

**Affiliations:** The Federal Research Center “Fundamentals of Biotechnology” of the Russian Academy of Sciences, 119071 Moscow, Russia; kulakova_masha95@mail.ru (M.K.); mari.pakhomova.99@mail.ru (M.P.); victoria.bidiuk@gmail.com (V.B.); e.alexey.mail@yandex.ru (A.E.); alexvir@gmail.com (A.A.)

**Keywords:** voltage-gated calcium channel, high-affinity Ca^2+^ uptake system, vacuolar Ca^2+^ ATPase, genetically encoded Ca^2+^ indicator, Hog1 phosphorylation, yeast, *Ogataea*

## Abstract

The cytosolic calcium concentration ([Ca^2+^]_cyt_) in yeast cells is maintained at a low level via the action of different transporters sequestrating these cations in the vacuole. Among them, the vacuolar Ca^2+^ ATPase Pmc1 crucially contributes to this process. Its inactivation in *Ogataea* yeasts was shown to cause sodium dodecyl sulfate (SDS) hypersensitivity that can be alleviated by the inactivation of the plasma membrane high-affinity Ca^2+^ channel Cch1. Here, we show that SDS at low concentrations induces a rapid influx of external Ca^2+^ into cells, while the plasma membrane remains impermeable for propidium iodide. The inactivation of Pmc1 disturbs efficient adaptation to this activity of SDS. The inactivation of Cch1 partially restores the ability of *pmc1* mutant cells to cope with an increased [Ca^2+^]_cyt_ that correlates with the suppression of SDS hypersensitivity. At the same time, Cch1 is unlikely to be directly involved in SDS-induced Ca^2+^ influx, since its inactivation does not decrease the amplitude of the rapid [Ca^2+^]_cyt_ elevation in the *pmc1-Δ* mutant. The obtained data suggest that the effects of *CCH1* inactivation on SDS sensitivity and coping with increased [Ca^2+^]_cyt_ are related to an additional Cch1 function beyond its direct involvement in Ca^2+^ transport.

## 1. Introduction

The response of eukaryotic cells to a number of external stimuli involves Ca^2+^ influx into the cytosol from the environment or internal depots. This requires maintaining a low cytosolic Ca^2+^ concentration ([Ca^2+^]_cyt_) and storage of these cations within these depots. At the same time, Ca^2+^ is required for processes taking place within the secretory organelles, which acquire it from the cytosol by the action of specific ion pumps [1]. In *Saccharomyces cerevisiae* yeast, plasma membrane proteins Cch1 and Mid1 represent the high-affinity Ca^2+^ uptake system (HACS) [2]. Cch1 was originally identified as a homolog of the voltage-gated Ca^2+^ ion channels of higher eukaryotes and was then shown to be involved in Ca^2+^ influx in response to treatment with alpha-factor mating pheromone [3,4,5]. Mid1 was identified as a protein required for survival upon alpha-factor treatment and involved in Ca^2+^ signaling [6]. Mid1 was shown to function as a mechanosensor when expressed in animal cells [7]. Mid1 and Cch1 interact with each other, and the inactivation of either one equally affects HACS function as well as cell survival during cell cycle arrest caused by the alpha-factor [3,8]. HACS is activated in response to insufficient Ca^2+^ supply to the secretory organelles independently of Cch1 and Mid1 expression [8]. An additional component of HACS, Ecm7, was identified by screening the collection of *S. cerevisiae* gene knock-out mutants for the abnormal uptake of Ca^2+^ after exposure to the mating pheromone or tunicamycin [9].

Alongside the high-affinity Ca^2+^ uptake system, low-affinity Ca^2+^ uptake has also been detected in yeast; however, only the *FIG1* gene has been shown to participate [10]. *Fig1* has also been shown to play a role in membrane fusion during cell mating [11].

The main yeast Ca^2+^ storage/sink organelle is the vacuole, which possesses its own Ca^2+^ transporters—ATPase Pmc1 and the H^+^/Ca^2+^ antiporter Vcx1 [12,13,14]. The Vcx1 antiporter is powered by the proton gradient across the vacuolar membrane, which is created by the vacuolar Vma proton pump [15]. Vcx1 and Pmc1 serve to replenish the vacuolar Ca^2+^ pool and to maintain cytosolic Ca^2+^ concentration at low levels. In *S. cerevisiae*, the inactivation of the *PMC1* gene leads to sensitivity to high concentrations of Ca^2+^ in culture medium and is lethal in the absence of the *PMR1* gene, which encodes a Ca^2+^/Mn^2+^ ion pump of the Golgi apparatus. Both these mutant phenotypes are suppressed by the inactivation of the Ca^2+^/calmodulin-dependent protein phosphatase calcineurin, indicating that an increased Ca^2+^ concentration blocks cell growth due to calcineurin activation, while Pmr1 acts together with Pmc1 in maintaining the cytosolic Ca^2+^ concentration at a low level [13]. Since in yeast, the vacuole is the main Ca^2+^ depot, its membrane possesses the Yvc1 channel, which releases Ca^2+^ into the cytosol [16]. Yvc1 is activated upon hyperosmotic shock due to its mechanosensitivity [17,18].

In methylotrophic *Ogataea* yeasts, the inactivation of Pmc1 also causes sensitivity to high Ca^2+^ levels in culture medium [19], but in contrast to *S. cerevisiae*, the growth defect in the *O. polymorpha pmc1-Δ pmr1-Δ* double mutant is due to insufficient Ca^2+^ supply to the secretory organelles [20]. Another trait that distinguishes *Ogataea pmc1-Δ* mutants from those of *S. cerevisiae* is hypersensitivity to sodium dodecyl sulfate (SDS) [19]. Hypersensitivity to detergents is usually observed in mutants with cell wall defects; however, no cell wall alterations have been detected in *Ogataea pmc1-Δ* mutants. Hypersensitivity to SDS in *S. cerevisiae* in some cases can also be related to Ca^2+^ homeostasis and signaling, since this phenotype of the *vps13-Δ* mutant can be suppressed by calcineurin inhibition or by an increased *PMC1* gene dosage [21,22]. *O. polymorpha pmc1-Δ* SDS hypersensitivity is suppressed by mutations of the Hog1 and Wee1 protein kinases involved in cell cycle regulation, as well as by the inactivation of the plasma membrane high-affinity Ca^2+^ channel Cch1 [19]. The latter suggests that the *pmc1-Δ* SDS sensitivity in *Ogataea* is caused by an elevated [Ca^2+^]_cyt_, while Cch1 inactivation decreases it. On the contrary, in this work, we show that SDS provokes Cch1-independent Ca^2+^ influx from the environment and that the SDS sensitivity of the *pmc1-Δ* mutant results from the inability to cope with the [Ca^2+^]_cyt_ increase, while Cch1 inactivation partially rescues this trait and improves the survival of the *pmc1-Δ* mutant during SDS treatment. This implies the presence of an additional function for Cch1 in Ca^2+^ homeostasis beyond transporting this ion through the plasma membrane.

## 2. Results

### 2.1. O. parapolymorpha Cells Survive in Presence of SDS at Concentrations Abolishing Cell Growth

Since SDS is a detergent, one can suggest that the inability of yeast cells to grow in the presence of this compound is due to the disruption of plasma membrane integrity. This should lead to rapid cell death accompanied by efficient staining with propidium iodide (PI). To study this, we first determined SDS’s minimal inhibitory concentration (MIC) for *O. parapolymorpha* strains bearing the GEM-GECO expression cassette. The strain with wild-type *PMC1* hardly grew at 0.008% and was unable to grow at 0.01% SDS, while the MIC for the *pmc1-Δ* mutant was between 0.001% and 0.0015%. The inactivation of *CCH1* improved the growth of the strain with wild-type *PMC1* at 0.008% and allowed the *pmc1-Δ* mutant to grow at 0.0015% SDS.

To test whether SDS causes rapid cell death, cells from stationary and logarithmic cultures were incubated with 0.01% SDS for 10 min or 1 h. In the logarithmic cultures, a similar reduction in the number of colony-forming units (CFU) was observed in all strains tested, despite the difference in the MIC (Figure 1). The survival rate in the stationary cultures was noticeably higher than that in logarithmic cultures in most cases (*p*-value < 0.05) except for the *pmc1-Δ* single mutant, indicating that dividing cells are more susceptible to SDS in low concentrations. Moreover, CFU numbers in the stationary cultures of the strains with wild-type *PMC1* were not significantly reduced even after 1 h incubation with 0.01% SDS. At the same time, a prolonged incubation with SDS led to a noticeable reduction in CFU numbers in the strains lacking this gene (Figure 1). CFU numbers in the *cch1-Δ* mutant were even increased after 10 min incubation with SDS. We suggest that SDS stimulates the separation of mother and daughter cells, which remain associated after the completion of cell division. This may lead to some overestimation of CFU numbers after SDS treatment. However, this suggestion has not been tested.

To study whether SDS makes cells more permeable to PI, we first assayed what incubation time is sufficient for the efficient staining of dead cells with this dye. To conduct this, cells, which were inactivated by incubation in a boiling water bath, were used. The shortest staining time we could achieve between adding the dye to the cell suspension and cytometric measurement was several seconds. This was sufficient to fully stain boiled cells, while longer incubation did not result in a noticeable alteration in the staining intensity. Thus, we concluded that the PI staining of dead cells does not depend on the duration of exposure to PI in the 1–10 min range. To study the permeability of living cells, cells from logarithmic YPD cultures were incubated in YPD containing 0.01% SDS for 50 min; then, PI was added, and cells were incubated 10 min more prior to flow cytometry. Only a minor fraction of cells incubated with SDS showed higher staining than untreated control cells (Figure 2). Notably, the staining intensity of this fraction was much lower than that of the cells, which were inactivated by boiling (Figure 2, black line) and most likely did not indicate cell death. Such a level of partial membrane permeabilization, which is not accompanied with cell death, was also demonstrated previously in *S. cerevisiae* during laser-induced cell printing [23] and treatment with copper ionophores and ergosterol synthesis inhibitors [24]. This indicated that SDS treatment does not make cells completely permeable to PI.

### 2.2. SDS Induces Ca^2+^ Entry into the Cytosol from the Environment

We suggested that SDS in low concentrations acts by affecting Ca^2+^ homeostasis. To test this, the genetically encoded fluorescent Ca^2+^ sensor GEM-GECO [25] was used. The maximum emission wavelengths of this sensor in the Ca^2+^-free and Ca^2+^-bound states are different, which allows for the monitoring of [Ca^2+^]_cyt_ by measuring the blue and green fluorescence intensity ratio (FL_450_/FL_525_) in single yeast cells using flow cytometry [26]. The K_d_ of the GEM-GECO complex with Ca^2+^ was previously determined as 340 nM [25], which allows for a rough collation of the FL_450_/FL_525_ values with the actual Ca^2+^ concentrations. In agreement with our suggestion, SDS caused a significant increase in [Ca^2+^]_cyt_ even in concentrations five-fold lower than the MIC. A further increase in the SDS concentration resulted in some increase in the [Ca^2+^]_cyt_ level (Figure 3). The rise in the FL_450_/FL_525_ ratio in response to SDS was even higher than that in response to treatment with 100 mM CaCl_2_ (Figure 3A,B). The [Ca^2+^]_cyt_ rise was due to Ca^2+^ influx from the environment, since the presence of chelating agents noticeably diminished this effect (Table 1, Figure 4).

### 2.3. The Ability of O. parapolymorpha Cells to Cope with the Elevated [Ca^2+^]_cyt_ Depends on Pmc1 and Cch1

Since Pmc1 is responsible for Ca^2+^ sequestration in the vacuole, its inactivation was expected to impair the cells’ ability to reduce [Ca^2+^]_cyt_ after the latter was elevated in response to SDS. The basal [Ca^2+^]_cyt_ level in the *pmc1-Δ* mutant was higher than that in the strain with wild-type *PMC1* (Table 1, Figure 4). After supplementing cultures with SDS at a final concentration of 0.01%, which is non-permissive for wild type cells, rapid [Ca^2+^]_cyt_ elevation was observed in all strains tested (Figure 3C,D). Like in the case of incubation with 100 mM CaCl_2_ (Figure 3B), the [Ca^2+^]_cyt_ rise, which occurred during the first few minutes of incubation with SDS, reached higher levels in the strains with wild-type *PMC1*. After reaching the peak or plateau, it went down in strains with wild-type *PMC1* or continued increasing in the strains lacking this gene. Notably, the dynamics of [Ca^2+^]_cyt_ in response to SDS depended on whether the exponentially growing or stationary phase culture was analyzed. Specifically, in the stationary cultures, the initial [Ca^2+^]_cyt_ rise was higher, and the following gradual [Ca^2+^]_cyt_ increase observed in the strains with *pmc1-Δ* mutation was steeper. The effect of *CCH1* inactivation in the *pmc1-Δ* mutant was also different: in the logarithmic culture, it mitigated [Ca^2+^]_cyt_ elevation during prolonged incubation with SDS, while in the stationary phase, its effect was the opposite (Figure 3C,D).

Reducing the SDS concentration to 0.004% allowed the exponentially growing *pmc1-Δ* cells to stabilize [Ca^2+^]_cyt_, while the loss of *CCH1* noticeably reduced the initial [Ca^2+^]_cyt_ elevation amplitude (Figure 3E). In the cells from the stationary culture, *CCH1* inactivation in the *pmc1-Δ* mutant increased the initial [Ca^2+^]_cyt_ elevation amplitude, but it slightly alleviated its further increase (Figure 3F).

Notably, the *cch1-Δ* mutation did not generally impair the rapid [Ca^2+^]_cyt_ elevation in response to SDS (slightly lowered or slightly boosted, depending on particular strain and culture growth phase), indicating that the SDS-induced Ca^2+^ influx does not require the Cch1 conduit. When cells were stressed by high external Ca^2+^ concentration, the inactivation of *CCH1* did not mitigate the [Ca^2+^]_cyt_ rise in the strain with wild-type *PMC1* and even boosted it in the *pmc1-Δ* mutant (Figure 3B). This demonstrates that the high-affinity Ca^2+^ channel is not involved in the Ca^2+^ influx at high external Ca^2+^ concentration. Although the inactivation of *CCH1* in the *pmc1-Δ* mutant led to a higher elevation of [Ca^2+^]_cyt_, it also partially restored the ability to cope with increased [Ca^2+^]_cyt_, since [Ca^2+^]_cyt_ had a tendency to decrease in the *pmc1-Δ cch1-Δ* double mutant, while it was practically constant in the strain bearing only the *pmc1-Δ* mutation (Figure 3B).

### 2.4. SDS Causes Transient Hog1 Phosphorylation

Since it was previously observed that the inactivation of the Hog1 protein kinase suppresses the *pmc1* mutant’s SDS hypersensitivity [19], we suggested that Hog1 is activated in response to SDS. Hog1 is the only representative of the p38 mitogen-activated protein kinase class in budding yeast, which is involved in response to hyperosmotic shock [27] and G2-M cell cycle transition control [28]. It is activated by the diphosphorylation of its C-terminal region, which is conservative throughout the different eukaryotic taxa. Thus, a monoclonal antibody against the diphosphorylated p38 C-terminal peptide was used to study Hog1 phosphorylation. As expected, hyperosmotic shock induced by 0.6 M NaCl led to effective Hog1 phosphorylation, which did not decrease for at least 20 min. SDS also led to an increase in Hog1 phosphorylation, though to a slightly lower level; however, this was not a long-lasting effect (Figure 5). Notably, the basal level of Hog1 phosphorylation as well as the response to SDS appeared to be lower in the *pmc1-Δ* mutant than in the wild-type strain (Figure 5, third panel).

## 3. Discussion

It was previously observed [19] that the inactivation of the vacuolar Ca^2+^ ATPase Pmc1 in *Ogataea* yeasts causes hypersensitivity to SDS, which is not accompanied by cell wall defects. This phenotype can be suppressed by mutations of the plasma membrane high-affinity Ca^2+^ ion channel Cch1, mitogen-activated protein kinase Hog1 or protein kinase Wee1 involved in the inhibition of the G2-M cell cycle transition. Besides functioning in response to high osmolarity shock [27], Hog1 is also involved in cell cycle control by phosphorylating the Hsl1 checkpoint kinase within the Hsl7-docking site, which results in Wee1 accumulation [28]. Taking all these facts into account, one can assume that cells can be more vulnerable to SDS during certain stages of the cell cycle. This is in agreement with our observations that most of the cells in stationary cultures remain alive in the presence of SDS at the MIC, while in the logarithmic cultures, this portion is noticeably reduced. At the same time, the inability of cells to form colonies after incubation with SDS was not accompanied by a substantial level of membrane permeabilization since almost no increase in PI staining was observed.

Using a genetically encoded Ca^2+^ indicator, we demonstrated that SDS causes a rapid [Ca^2+^]_cyt_ rise, while the plasma membrane retains a considerable barrier function towards propidium iodide. The influx of Ca^2+^ leads to temporary Hog1 phosphorylation, which rapidly returns to baseline levels. This might suggest that the loss of Pmc1 leads to SDS sensitivity by activating Hog1 due to the increased [Ca^2+^]_cyt_; however, the short duration of Hog1 phosphorylation is hardly consistent with this. Moreover, after the SDS-induced rise, the Hog1 phosphorylation declined in the *pmc1-Δ* mutant faster than in wild-type cells, despite the increased [Ca^2+^]_cyt_. We suggest that the basal level of Hog1 phosphorylation is sufficient to fulfill its function in cell cycle control, while the increased [Ca^2+^]_cyt_ affects this pathway downstream of Hog1. For example, this can be due to a Ca^2+^-dependent increase in Wee1 activity, since it was shown in *S. cerevisiae* that calcineurin and Mpk1 regulate Wee1 activation at the transcriptional and post-translational levels, respectively, and both are required for the calcium-induced delay in the G2 phase [29].

Since Cch1 is a pore-forming component of HACS, one could suggest that the suppression of the *pmc1-Δ* mutant phenotype by Cch1 inactivation comes from the decrease in [Ca^2+^]_cyt_ due to the defect of the Ca^2+^ transport through the plasma membrane. However, this suggestion is not supported by the results obtained in this work. When *CCH1* was inactivated, we observed some decrease in the SDS-induced [Ca^2+^]_cyt_ rise in the strain with wild-type *PMC1* and a decrease in the basal [Ca^2+^]_cyt_ in the *pmc1-Δ* mutant. However the amplitude of the SDS-induced [Ca^2+^]_cyt_ rise in the *pmc1-Δ* mutant was not decreased. At the same time, the loss of Cch1 improved the ability of the *pmc1-Δ* mutant to cope with the increased [Ca^2+^]_cyt_ in cells from exponentially growing cultures. The survival rate in stationary phase cells of the *pmc1-Δ* mutant was also increased when the *CCH1* gene was inactivated despite the increased [Ca^2+^]_cyt_. This may indicate a regulatory role of Cch1 in Ca^2+^ homeostasis and/or in response to the increased [Ca^2+^]_cyt_.

## 4. Materials and Methods

### 4.1. Yeast Strains

The strains used in this study are listed in Table 2. The process of obtaining the *O. parapolymorpha* DL5 strain bearing the GEM-GECO expression cassette, the *pmc1-Δ* mutant DL5-pmc1, their prototrophic derivatives and the *hog1-Δ* mutant was described in [26]. The *CCH1* gene was disrupted in the DL5 and DL5-pmc1 strains by transformation with a disruption cassette, which is described elsewhere [19].

### 4.2. Culture Media and Yeast Transformation

Complex media contained 2% Peptone, 1% yeast extract and 2% glucose (YPD) or 1% sucrose (YP-Suc) as a carbon source. The synthetic medium SC-D (2% glucose, 0.67% Yeast Nitrogen Base with ammonium sulfate) was used for the selection of transformants. Yeast cells were transformed using the Li-acetate method [30] with some modifications [31].

### 4.3. Monitoring of Cytosolic Ca^2+^ Concentration

The GEM-GECO genetically encoded Ca^2+^ indicator [25] was used to monitor [Ca^2+^]_cyt_ in individual *O. parapolymorpha* cells using flow cytometry as described previously [26]. To conduct this, *O. parapolymorpha* strains possessing the GEM-GECO-encoding gene under the control of the *MAL1* promoter were used. This promoter is repressed during glucose consumption and induced when cells utilize some disaccharides including sucrose. Thus, to obtain stationary-phase cells expressing GEM-GECO, overnight YPD cultures were diluted 50-fold with YP-Suc and incubated overnight. To obtain logarithmic-phase cells, the overnight YP-Suc cultures were diluted 50- (in case of strains with wild-type *PMC1*) or 25-fold (in case of *pmc1-Δ* mutants) with fresh YP-Suc and incubated for 4 h. Fluorescence excited by a 488 nm laser and measured using 525/40 nm bandpass filters (“FITC channel”) was used to calculate autofluorescence, since it was previously found to not depend on the presence of GEM-GECO and showed a good correlation with 405 nm-excited 525 nm and 450 nm autofluorescence [26]. To conduct this, trend lines of the dependence of autofluorescence values at 525 nm and 450 nm in “FITC channel” fluorescence in strains lacking GEM-GECO were calculated using MS Excel (Microsoft Office 2021 Professional Plus). Based on the obtained functions and “FITC channel” fluorescence values in the strains producing GEM-GECO, the values of autofluorescence were calculated for individual cells and subtracted from the measured 525 nm and 450 nm fluorescence values. The obtained values were normalized to autofluorescence at each wavelength. The FL_450_/FL_525_ ratio was used as a characteristic of [Ca^2+^]_cyt_ [25].

### 4.4. Growth Inhibition and Cell Survival Assays

Cells were grown in liquid YPD for 16–20 h at 37 °C to obtain stationary cultures, which were then diluted 50-fold with fresh YPD and incubated in a shaker–thermostat at 37 °C for 3.5–4 h to obtain logarithmic cultures.

To determine SDS’s minimal inhibition concentrations (MICs), stationary or logarithmic YPD cultures were diluted 200- (stationary) or 50- (logarithmic) fold with fresh YPD and mixed with equal volumes of YPD supplemented with different SDS concentrations in a 96-well plate. The obtained cell suspensions were incubated overnight in a plate shaker at 37 °C to allow cells to grow. The ability of the cells to grow was assessed visually by culture turbidity.

To assay the survival rate upon SDS treatment, the stationary and logarithmic cultures were diluted with YPD supplemented with 0.01% SDS to OD_600_ = 0.1. Cells from suspensions after 10 and 60 min incubation, as well as cells which were not incubated with SDS (control), were collected by centrifugation, washed with fresh YPD and spread onto YPD plates in appropriate dilutions to obtain distinct colonies. The survival rate was calculated as the ratio between the number of colony-forming units in the suspensions of the SDS treated and control cells.

### 4.5. Analysis of PI Staining Intensity

Logarithmic YPD cultures were diluted with YPD supplemented with 0.01% SDS to OD_600_ 0.1 and incubated for 50 min; then, 1 μg/mL PI was added, and then suspensions were incubated for 10 min. The control cells were incubated in YPD with 1 μg/mL PI and without SDS for 10′. Cells, which were inactivated by 1′ incubation in a boiling water bath, were used as a completely PI-permeable reference. PI fluorescence in individual cells was analyzed by flow cytometry (488 nm excitation laser, 690/50 nm bandpass filter).

### 4.6. Analysis of Hog1 Phosphorylation

The alterations in Hog1 phosphorylation were assayed by immunoblotting with anti-Phospho-p38 monoclonal antibody (Cell Signaling Technology, Inc., Danvers, MA, USA, cat. #9215). Total Hog1 content was assayed by immunoblotting with polyclonal rabbit antisera against *Escherichia coli*-expressed *O. polymorpha* Hog1. In preliminary experiments, we observed that the phosphorylated Hog1 can undergo dephosphorylation during the preparation of cell lysates from live cells. To avoid this, cells were fixed by the rapid mixing of liquid yeast cultures with a 2.4 volume of ice-cold methanol and overnight incubation on ice. Cells were harvested by centrifugation and kept at room temperature in open tubes for 30 min to let the residual methanol evaporate. Then cells were washed twice with water and divided into two portions, one of which was used to prepare SDS electrophoresis samples using alkaline treatment as described elsewhere [32], while another one was used to assay total cellular protein for insuring an equal amount of protein in each lane. The latter was conducted according to the previously described protocol [33] as follows. Cells were suspended in 0.7 mL of water, mixed with 0.35 mL of 3M NaOH and incubated in a boiling water bath for 5 min. Then, 0.35 mL of 2.5% CuSO_4_·5H_2_O and the mix was incubated for 5 min at room temperature. The pellet was removed by centrifugation in a microcentrifuge at the maximum speed for 5 min. The optical densities of the supernatants were measured at 555 nm wavelength. Solutions of bovine serum albumin (0.2–2 mg/mL) were used as standards. Volumes of the electrophoresis samples were adjusted to obtain the same protein concentrations.

## Figures and Tables

**Figure 1 ijms-25-11450-f001:**
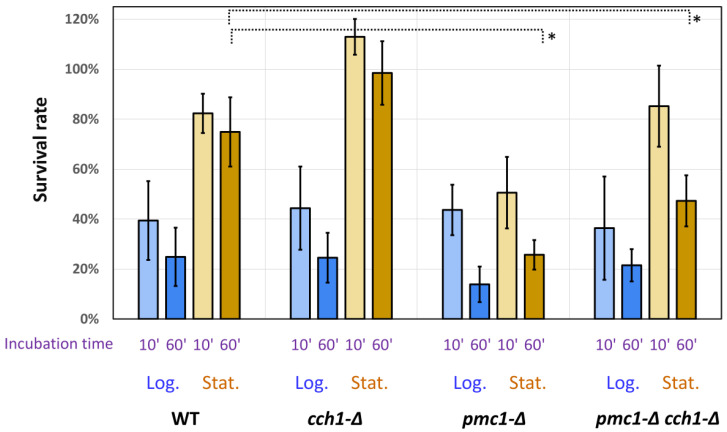
Change in CFU numbers (survival rate) after 10′ and 60′ incubation in presence of 0.01% SDS. Log., logarithmic cultures; Stat., stationary cultures; WT, DL5-LC strain; *cch1-Δ*, DL5-cch1 strain; *pmc1-Δ*, DL5-pmc1-LC strain; *pmc1-Δ cch1-Δ*, DL5-pmc1-cch1 strain. For details, see Section 4. *—*p*-value < 0.005.

**Figure 2 ijms-25-11450-f002:**
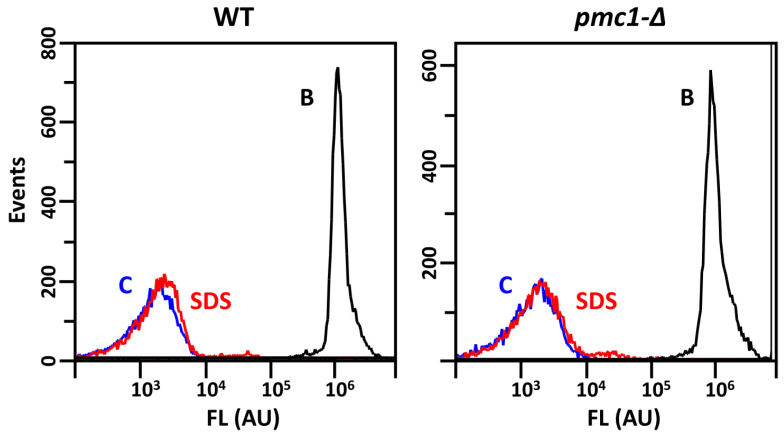
Distribution of fluorescence (FL) of DL5-LC (WT) and DL5-pmc1-LC (*pmc1-Δ*) cells after PI staining. Red line “SDS”, cells after 1 h incubation with 0.01% SDS; blue line “C”, untreated control cells; black line “B”, cells inactivated by boiling. Comparison of three replicates is represented in Appendix A.

**Figure 3 ijms-25-11450-f003:**
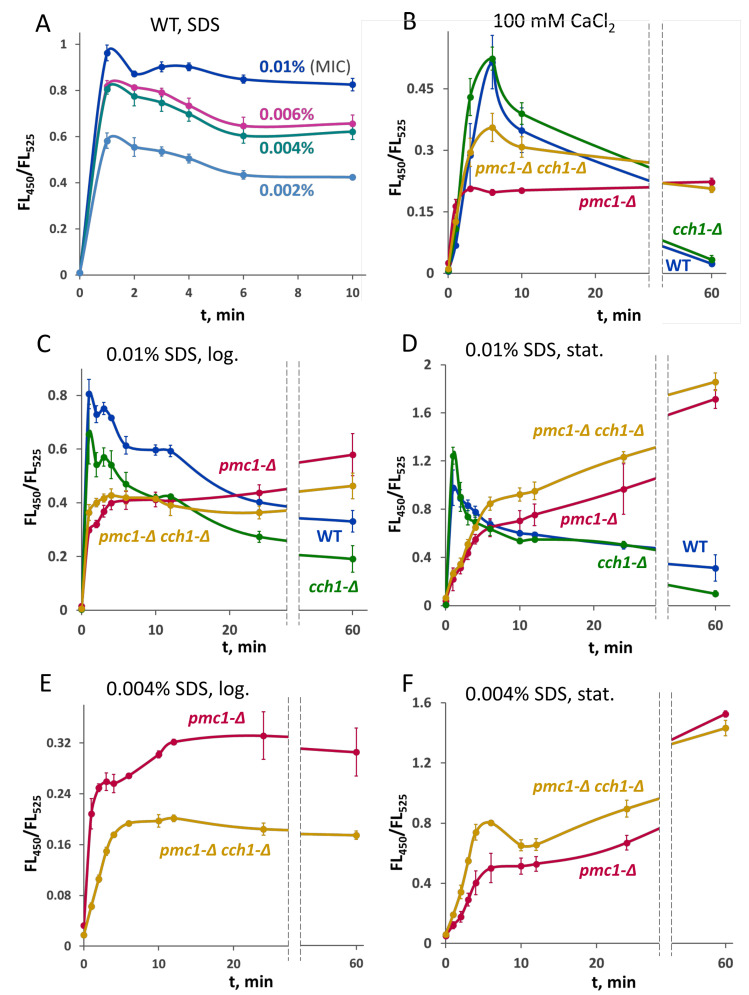
Dynamics of median FL_450_/FL_525_ values of cells of DL5-LC (WT), DL5-cch1 (*cch1-Δ*), DL5-pmc1-LC (*pmc1-Δ*) and DL5-pmc1-cch1 strains expressing GEM-GECO. (**A**), response of DL5-LC cells in logarithmic culture to different SDS concentrations; (**B**), response of cells in logarithmic cultures to 100 mM CaCl_2_; (**C**), response of cells in logarithmic cultures to 0.01% SDS; (**D**), response of cells in stationary cultures to 0.01% SDS; (**E**), response of cells in logarithmic cultures to 0.004% SDS; (**F**), response of cells in stationary cultures to 0.004% SDS. Median FL_450_/FL_525_ values of 10^4^ cells obtained from 3 replicates were averaged, and standard deviations were calculated.

**Figure 4 ijms-25-11450-f004:**
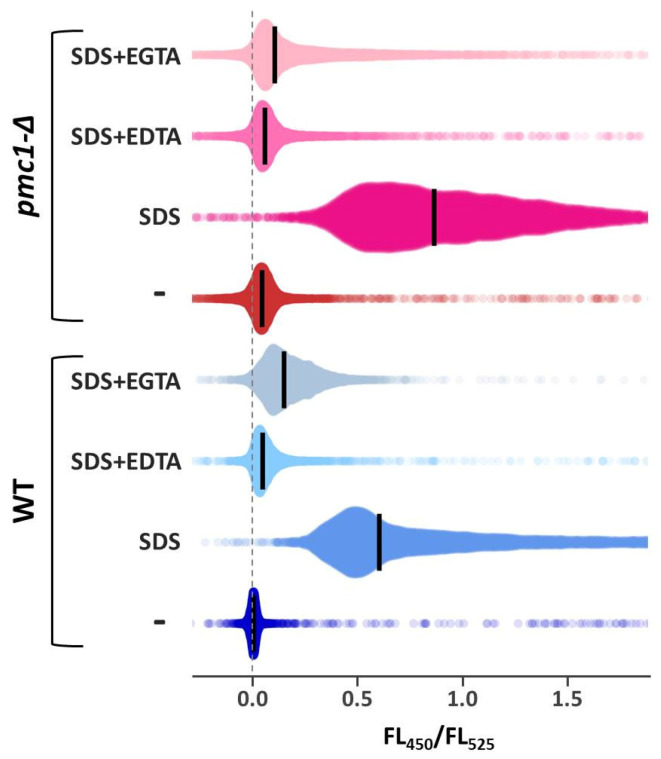
Distributions of FL_450_/FL_525_ in exponentially grown cultures of DL5-LC (WT) and DL5-pmc1-LC (*pmc1-Δ*) strains before (-) or after 10′ incubation with 0.01% SDS in absence (SDS) or presence of 10 mM EDTA (SDS + EDTA) or 20 mM EGTA (SDS + EGTA). Median values are indicated by bars.

**Figure 5 ijms-25-11450-f005:**
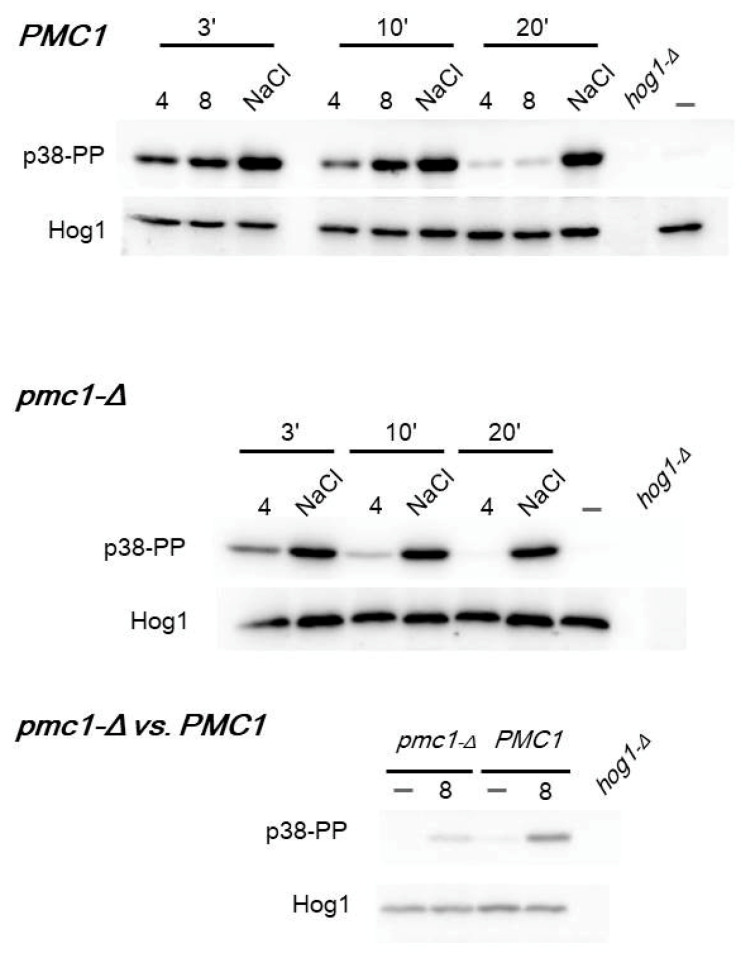
SDS-PAGE and immunoblotting of phosphorylated (p38PP) and total Hog1 (Hog1). Exponential cultures were supplemented with 0.004% SDS (4), 0.008% SDS (8) or 0.6 M NaCl and incubated for 3, 10 or 20 min. Untreated cells (-) were used as reference. *hog1-Δ* mutant was used as negative control. Panel “*PMC1*”—DL5-LC strain; panel “*pmc1-Δ*”—DL5-pmc1-LC strain; panel “*pmc1-Δ* vs. *PMC1*”—comparison of DL5-pmc1-LC and DL5-LC strains after 20 min incubation with SDS.

**Table 1 ijms-25-11450-t001:** Median FL_450_/FL_525_ values * in individual cells of DL5 (WT) and DL5-pmc1 (*pmc1-Δ*) strains after 10 min incubation of logarithmically grown cultures in presence of 0.01% SDS alone or in combination with 10 mM ethylenediaminetetraacetate (EDTA) or 20 mM ethylene glycol-bis(β-aminoethyl ether)-N,N,N′,N′-tetraacetate (EGTA).

Supplement	WT	*pmc1-Δ*
-	0.0075 ± 0.0006	0.046 ± 0.003
SDS	0.65 ± 0.04	0.86 ± 0.01
SDS + EDTA	0.049 ± 0.003	0.061 ± 0.001
SDS + EGTA	0.15 ± 0.01	0.110 ± 0.005

* The median FL_450_/FL_525_ values of 10^4^ cells obtained from 3 replicates were averaged, and standard deviations were calculated.

**Table 2 ijms-25-11450-t002:** *O. parapolymorpha* strains used in this study.

Strain	Genotype *
DL5	*leu2 {P_MAL1_-GEM-GECO}*
DL5-LC	*leu2 {P_MAL1_-GEM-GECO} {LEU2}*
DL5-cch1	*leu2 cch1::LEU2 {PMAL1-GEM-GECO}*
DL5-pmc1-LC	*leu2 pmc1::loxP {P_MAL1_-GEM-GECO} {LEU2}*
DL5-pmc1-cch1	*leu2 pmc1::loxP cch1::LEU2 {P_MAL1_-GEM-GECO}*
DL5-hog1	*leu2 hog1::LEU2 {P_MAL1_-GEM-GECO}*

* Names of genes integrated into unidentified genome loci as part of plasmid are shown in brackets.

## Data Availability

The original contributions presented in the study are included in this article; further inquiries can be directed to the corresponding author.

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
