# Peer review of "High-Affinity Plasma Membrane Ca2+ Channel Cch1 Modulates Adaptation to Sodium Dodecyl Sulfate-Triggered Rise in Cytosolic Ca2+ Concentration in Ogataea parapolymorpha"

_ijms, 2024, doi:10.3390/ijms252111450_

Round 1
Reviewer 1 Report
Comments and Suggestions for Authors
In this paper, the authors address the functions of the plasma membrane Ca2+ channel (Cch1) and vacuolar Ca2+ pump (Pmc1) in Ca2+ homeostasis, and specifically a rapid Ca2+ influx at low concentrations of SDS. The authors previously showed that sublethal concentrations of SDS provoke an influx of Ca2+. Consistent with this, in O. polymorpha, a pmc1∆ mutant is hypersensitive to SDS, but this sensitivity can be suppressed by mutation of Cch1. The major conclusion from the data is that the effects of Cch1 deletion on SDS hypersensitivity is related to a function of Cch1 beyond Ca2+ transport. However, the data do not fully support this conclusion.
There is no question that there is an influx of Ca2+ from outside the cell with SDS treatment, as Ca2+ chelators appear to reduce cytosolic Ca2+ levels in both wild-type and pmc1∆ mutants (Table 1). However, a major part of the argument for an additional function of Cch1 is based on the high levels of cytosolic Ca2+ in a pmc1∆ cch1∆ double mutant, which continue to rise with time. The tight focus on these two players in Ca2+ homeostasis does not take into account what is happening with other important players, such as vacuolar Vcx1 and calcineurin. Calcineurin would be highly activated under the conditions of high cytosolic Ca2+ seen here and could affect many functions. It is premature to conclude that Cch1 has a new function without considering Ca2+ homeostasis more broadly.
In addition, the authors suggest that there is no increase in plasma membrane permeability because PI staining does not increase in the pmc1∆ mutant after 10 min. treatment with 0.01% SDS. However, it is notable that PI staining is higher in the pmc1∆ mutant than in wild-type without SDS and comparable to the wild-type signal with SDS in Figure 2. This suggests that there could be an increase in plasma membrane permeability in the mutant that could contribute to the high sustained levels of cytosolic Ca2+.
Other points to consider:
1). It would be helpful to do a calibration of FL450/FL525 to Ca2+ concentration (by collapsing the Ca2+ gradients and measuring the ratio at different Ca2+ concentrations). This would give a clearer insight into how much cytosolic Ca2+ concentrations are really increasing. It would also be helpful to discuss in the Methods how the correction for autofluorescence is done in more detail, since some of the strains may be quite sick.
2). Figure 1 needs more explanation--a legend for the y-axis, some statistical comparisons, and an explanation of >100% survival.
Author Response
Q1: In this paper, the authors address the functions of the plasma membrane Ca2+ channel (Cch1) and vacuolar Ca2+ pump (Pmc1) in Ca2+ homeostasis, and specifically a rapid Ca2+ influx at low concentrations of SDS. The authors previously showed that sublethal concentrations of SDS provoke an influx of Ca2+. Consistent with this, in O. polymorpha, a pmc1∆ mutant is hypersensitive to SDS, but this sensitivity can be suppressed by mutation of Cch1. The major conclusion from the data is that the effects of Cch1 deletion on SDS hypersensitivity is related to a function of Cch1 beyond Ca2+ transport. However, the data do not fully support this conclusion.
A: Actually the effect of SDS on the cytosolic Ca2+ concentration is described here for the first time. This is one of the manuscript’s conclusions. Previously, we have observed that pmc1 mutant is hypersensitive to SDS and that this phenotype can be suppressed by inactivation of CCH1 or HOG1. That time we did not have a proper instrument to assess cytosolic calcium concentration in Ogataea cells. Fortunately, the GEM-GECO indicator had been described (doi:10.1126/science.1208592) and, after testing its efficacy and limitations in our previous work (doi:10.3390/ijms231710004), we could apply it for this purpose now.
Q2: There is no question that there is an influx of Ca2+ from outside the cell with SDS treatment, as Ca2+ chelators appear to reduce cytosolic Ca2+ levels in both wild-type and pmc1∆ mutants (Table 1). However, a major part of the argument for an additional function of Cch1 is based on the high levels of cytosolic Ca2+ in a pmc1∆ cch1∆ double mutant, which continue to rise with time. The tight focus on these two players in Ca2+ homeostasis does not take into account what is happening with other important players, such as vacuolar Vcx1 and calcineurin. Calcineurin would be highly activated under the conditions of high cytosolic Ca2+ seen here and could affect many functions. It is premature to conclude that Cch1 has a new function without considering Ca2+ homeostasis more broadly.
A: We disagree with this concern. The question was: is Cch1 the major gate for the Ca2+ influx in response to SDS? In this case its inactivation would strongly decrease the Ca2+ influx in all cases. However we observed only modest effect of CCH1 inactivation on the initial Ca2+ rise and this effect could be positive or negative depending on strain genotype and conditions. This is not consistent with the idea that SDS induces Ca2+ influx through the Cch1 channel. At the same time, inactivation of CCH1 suppresses SDS sensitivity and affects cytosolic [Ca2+] dynamics after SDS treatment. Possibly, Vcx1 and calcineurin are involved in these effects, however this does not argue against the conclusion that Cch1 is not the main gate for the SDS-induced Ca2+ influx. Thus the observed effects prove that the role of Cch1 is not solely that of a Ca2+ influx channel and suggest the presence of an additional Cch1 function. We hope to study whether Vcx1 or calcineurin are related to this function in future.
Q3: In addition, the authors suggest that there is no increase in plasma membrane permeability because PI staining does not increase in the pmc1∆ mutant after 10 min. treatment with 0.01% SDS. However, it is notable that PI staining is higher in the pmc1∆ mutant than in wild-type without SDS and comparable to the wild-type signal with SDS in Figure 2. This suggests that there could be an increase in plasma membrane permeability in the mutant that could contribute to the high sustained levels of cytosolic Ca2+.
A: This is probably due to imperfectness of the flow cytometer measurements. We inspected all of the replicates and did not see this difference between peaks before and after SDS treatment in the wild-type strain. This is a usual problem with flow cytometry. Now we placed the data from another replicate into the figure, while comparison of results of three replicates is presented as a supplementary figure. The PI staining in WT and pmc1-Δ mutant also does not show a difference in other replicates. We thank the reviewer for noticing this irregularity.
Q4: It would be helpful to do a calibration of FL450/FL525 to Ca2+ concentration (by collapsing the Ca2+ gradients and measuring the ratio at different Ca2+ concentrations). This would give a clearer insight into how much cytosolic Ca2+ concentrations are really increasing.
A: This is not as a trivial task as it may seem. We tried to use two different ionophores A23187 and ionomycin. The former one induces additional cell autofluorescence, which significantly distorts GEM-GECO analysis. The latter one’s effect on the cytosolic [Ca2+] was much less pronounced than the effect of SDS indicating that this ionophore is not potent enough in Ogataea to perform such experiments. At the same time, the conclusions we come to in this work do not require precise assessment of Ca2+ concentration. To give some insight into the collation of FL450/FL525 values with real Ca2+ concentrations, now we mentioned that previously determined GEM-GECO-Ca2+ Kd is 340 nM. Maximum FL450/FL525 values observed in our experiments were approximately 2-2.5. It is reasonable to expect that approximately half of this value corresponds to 340 nM. This is a very low concentration (more than 1000 times lower than that in culture medium), which is hardly achievable since cells should retain some calcium in the vacuole and ER. That is why we prefer to rely on the GEM-GECO kinetic characteristics determined earlier.
Q5: It would also be helpful to discuss in the Methods how the correction for autofluorescence is done in more detail, since some of the strains may be quite sick.
A: This information has been added. Autofluorescence was assessed for each mutant in all experiments.
Q6: Figure 1 needs more explanation--a legend for the y-axis, some statistical comparisons, and an explanation of >100% survival.
A: The figure has been modified. P-values were indicated in the figure legend or in the text. We suggest that SDS can stimulate separation of mother and daughter cells which remain associated after completion of cell division that may lead to some over estimation of CFU after SDS treatment. This is discussed in new version of the manuscript.

Reviewer 2 Report
Comments and Suggestions for Authors
Comments in pdf file

Author Response
Q1: I am not sure that we can write about adaptation to increased cytosolic Ca2+ concentration. This was studied in the presence of SDS, so in special stress conditions.
A: Not only SDS was used to stimulate cytosolic [Ca2+ ] rise, but elevated external Ca2+ concentration also. In all cases we observed effects of Cch1 inactivation on ability of cells to cope with increased cytosolic [Ca2+]. To our opinion, the title mirrors this fact. Some other results obtained in this work, which we think are also important, are not mentioned in the title because one can only focus on a limited number of points in the title. However, we agree with the reviewer that SDS and even high external Ca2+ concentration are quite specific stimuli and we mentioned SDS in the revised title.
Q2: What is the statistical significance of the changes for the values shown in Figure 1 and Figure 3. This is an important question when we want to draw conclusions about the role of the Cch1 protein. In particular, I find the results for the cch1 mutant shown in Figure 1 difficult to interpret. How is it possible that more than 100% of the cells survive? How did treatment with 10' SDS result in more growing colonies compared to untreated cells in the cch1 mutant?
A: p-Values are added to the text and the figure legend. The increased CFU number after SDS treatment is discussed now.
Q3: Also by combining the data in Figure 1 with those in Figure 3C and D, it can be said that young cells of the strains tested survive the stress of treatment with 0.01% SDS in a comparable manner and this is not correlated with the level of calcium in the cytoplasm. On the other hand, when the cells are in stationary phase all the cells tested except the pmc1 mutant survive the SDS stress better and to say something about the correlation with calcium levels it seems to have it lower than the better growing pmc1 cch1 mutant (but statistical significance is necessary). Where does this extra calcium come from according to the authors?
A: The reviewer is correct that there is no good correlation between Ca2+ levels and survival rate when we compare stationary and exponentially growing cells. It is also true that compared to pmc1, the pmc1 cch1 strain exhibits a higher level of [Ca2+ ]cyt in the stationary cultures after supplementing with SDS. At the same time Cch1 inactivation lowers [Ca2+]cyt concentration during prolonged incubation, which is visible when lower SDS concentration was used (Figure 3F). We cannot readily explain the effects observed by the reviewer with the current data, however they do not seem to argue against, but rather for our statement that Cch1 is not acting as a gate for calcium ions in response to SDS.
Q4: What do the authors mean when they comment that Cch1 has a regulatory role in Ca2+ homeostasis (line258); i.e. what does Cch1 do? Once these doubts have been cleared up, one can reflect on the title.
A: Regarding the regulatory role of Cch1, this is just our interpretation of the obtained results and it is expressed as a suggestion. The title says that Cch1 modulates adaptation to increased [Ca2+ ]cyt. This is just a short description of the results we obtained here and there is no claim on the mechanisms involved in this adaptation. While we do not want to speculate too much about this in the paper, our main hypotheses would be that Cch1 senses cytosolic Ca2+ level and activates the low affinity Ca2+ uptake system and/or represses Ca2+ sequestration machinery (e.g. Pmc1, Vcx1, Pmr1). Thus its absence may make the cell more prepared for coping with increased [Ca2+]cyt. Such regulation may compete with the Crz1-dependent regulation that should make the effects more complex. This can be a topic for the future research.
Q5: The authors write that the growth inhibitory effect of calcium ions in the cytoplasm is due to the activation of calcineurin. Does the application of FK506 have a protective effect on the pmc1 mutant and the cells start to divide despite SDS? In S.cerevisiae in the vps13 mutant, sensitivity to SDS is suppressed by FK506, genetic restriction of calcineurin activity by overexpression of gene (RCN2) encoding its inhibitor, but also by overexpression of PMC1.
A: The detrimental effect of increased cytosolic [Ca2+] is not our idea. That was published previously and we refer to this work in Introduction (reference 13). The studies indicated by the reviewer also support this idea. Unfortunately we missed the corresponding articles when we were writing the manuscript. Now they are mentioned in Introduction. At the same time, studying involvement of calcineurin was not the task of the work described in the manuscript, but we appreciate this suggestion and will study this in the future.
While we could not get quick access to FK506, we tested whether application of another inhibitor of calcineurin function has an effect on SDS sensitivity in the pmc1-Δ mutant. Indeed, the presence of cyclosporine suppressed the SDS sensitivity. However, we also found cases in the literature where some cyclosporine effects were found to be calcineurin independent. So we are hesitant to include these data without a closer look into what is going on. We thank the reviewer for suggesting this experiment and hope to get to the bottom of it in our future work.
Q6: Add reference to text that the pmc1 mutant of S. cerevisiae is not sensitive to SDS.
A: The reference has been added.
Q7: What is FL(AU)?
The description of the method should be in the materials and methods, not in the figure caption.
A: FL means fluorescence. This is explained in the figure legend now. AU is standard notation for arbitrary units. The description has been moved to Materials and Methods.
Q8: Table 1
Why did the authors give the median and not the mean value? I can understand giving the median if there are many values that do not have a normal distribution. In this case it is the middle value of the three. Please explain this.
A: This is a misunderstanding. These are mean values of medians (medians from 3 replicates were averaged). The footnote was rephrased. We hope it is now more comprehensive.
Q9. Figure3B
Cyrillic lettering
line 215
0.012% SDS (12) delete, there is no such thing in the figure
Line 303
Cells were rather collected
A: These items were corrected.
Q10: Methods
Line 299
What was after the overnight incubation, what was measured and how?
A: The explanation was added